# Tracing Water–Rock–Gas Reactions in Shallow Productive Mud Chambers of Active Mud Volcanoes in the Caspian Sea Region (Azerbaijan)

Aygun Bayramova [1], Orhan R. Abbasov [1], Adil A. Aliyev [1], Elnur E. Baloglanov [1], Franziska M. Stamm [2], Martin Dietzel [2] and Andre Baldermann [2,*]

1 Mud Volcanism Department, Institute of Geology and Geophysics, Ministry of Science and Education of the Republic of Azerbaijan, H. Javid Av., 119, Baku AZ1143, Azerbaijan; bayramova.aygun97@gmail.com (A.B.); ortal80@bk.ru (O.R.A.); ad_aliyev@mail.ru (A.A.A.); elnur1001@mail.ru (E.E.B.)
2 Institute of Applied Geosciences & NAWI Graz Geocenter, Graz University of Technology, Rechbauerstraße 12, 8010 Graz, Austria; fstamm@tugraz.at (F.M.S.); martin.dietzel@tugraz.at (M.D.)
* Correspondence: baldermann@tugraz.at; Tel.: +43-(0)316-873-6850

**Abstract:** We present geochemical and mineralogical datasets for five new mud volcanoes in continental Azerbaijan (Hamamdagh and Bendovan) and the adjacent Caspian Sea (Khara-Zire, Garasu and Sangi-Mughan). The fluid ejects have a Na–Cl-type composition and are generated by the mixing of evaporated Caspian seawater and low- to high-salinity pore waters, as indicated by Br–B and Cl–B systematics and Na–K and $SiO_2$ geo-thermometers. The fluids contain high concentrations of As, Ba, Cu, Si, Li, Sr and Zn (60 to 26,300 ppm), which are caused by surface evaporation, pyrite oxidation, ion exchange reactions and hydrocarbon maturation in Oligocene-Miocene 'Maykop' shales. The solid ejects comprise liquid, oily and brecciated mud, mud/claystones and sandstones. The mud heterogeneity of the volcanoes is related to the geological age and different sedimentological strata of the host rocks that the mud volcanoes pass through during their ascent. All ejects show evidence of chemical alterations via water–rock–gas reactions, such as feldspar weathering, smectite illitization and the precipitation of Fe-(hydr)oxides, calcite, calcian dolomite, kaolinite and smectite. The studied localities have petrographic similarities to northern extending mud volcano systems located on Bahar and Zenbil islands, which suggests that mud volcanoes in the Caspian Sea region are sourced from giant shallow mud chambers (~1–4 km depth) located in Productive Series strata. Our results document the complex architecture of the South Caspian Basin—the most prolific hydrocarbon region in the world.

**Keywords:** mud volcanism; seep water chemistry; water-rock interaction; mud mineralogy; mud chemistry; Caspian Sea; Azerbaijan; Productive Series; Maykop shales





## 1. Introduction

Mud volcanoes provide a unique window to the architecture of sedimentary basins, convergent margins and subduction zones [1,2]. They produce scenic landscapes and morphological features on the Earth's surface, such as gryphons, mud and scoria cones, salses, springs and fire columns [3–6]. However, the release of toxic and radioactive elements (arsenic, As; barium, Ba; copper, Cu; strontium, Sr; uranium, U; zinc, Zn, etc.) and (a) biogenic gases (e.g., methane, $CH_4$ and minor carbon dioxide, $CO_2$) during the active and dormant periods of volcanoes can impact the chemistry of local surface water, groundwater, interstitial solutions and sediments, as well as the Earth's atmospheric gas budget [7–16]. In many places, mud volcanic activity is triggered by the rapid subsidence of soft, fine-grained sediments rich in water and organic matter in a compressive tectonic setting, which promotes the generation of hydrocarbons and, subsequently, the formation

of fluid–mud–gas mixtures migrating upward along fractures and faults in the compacted host rocks under overpressure [1,17,18].

Mud volcanoes are pervasive within the Caspian Sea region, particularly in Azerbaijan, which is a transcontinental country located at the boundary between Eastern Europe and Western Asia (Figure 1a). Thus far, 353 mud volcanoes have been identified in East Azerbaijan and in the shallow-water area of the South Caspian Sea, both onshore (199) and offshore (154) [3,19–21]. Mud volcanism in this area was caused by (i) fast sedimentation rates of up to 2.4 km per million years in the Pliocene, (ii) massive $CH_4$ production in the Miocene–Pliocene 'Productive Series' sediments and in the Oligocene–Miocene 'Maykop' shales, (iii) the presence of major folding zones and (iv) the tectonic compression of the South Caspian Basin [14,22,23]. The latter causes the generation of low-density fluid–mud–gas mixes from ~1 to 8 km in depth [5], which migrate upward in the compacted, fractured sedimentary rocks [1]. The resulting mud diapirism ejects clay-rich sediments that form the Earth's landscape, in addition to escaping fluid and gas phases [5,7,13,15,24]. However, the age of the basement, the beginning and progression of mud diapirism and the origin of the South Caspian Basin are disputed, with most published models postulating an initiation between the Jurassic and Miocene [25].

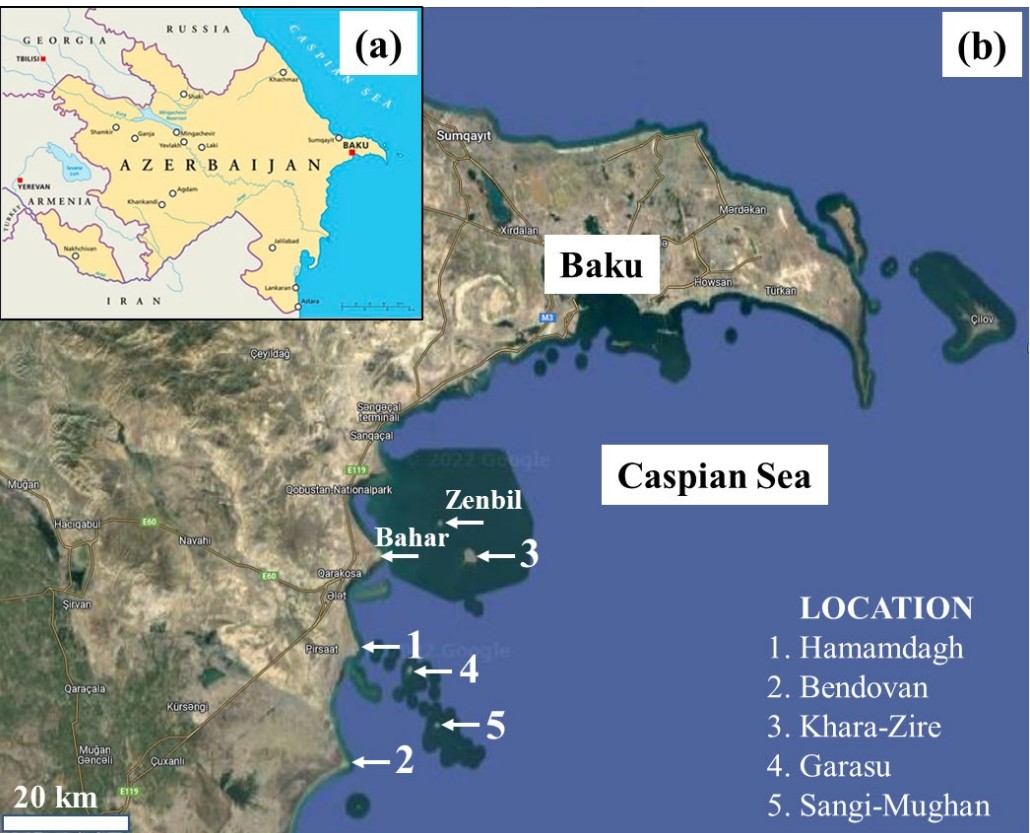

**Figure 1.** (**a**) Location map of Azerbaijan in Western Asia. (**b**) Google Earth view showing the location of the investigated mud volcanoes in the Baku area (onshore sites: 1 and 2) and on islands within the Caspian Sea (offshore sites: 3–5), Azerbaijan.

The widespread, active and passive mud volcano systems located onshore in East Azerbaijan are relatively well documented with respect to their fluid and solid ejects [4,11,26–28]. On the contrary, ejects of island mud volcanoes are much less well characterized. This is because conventional sampling is difficult due to the isolated location of these islands in the Caspian Sea. For instance, more than eight new islands formed within the Baku archipelago and the Caspian Sea region as a result of (sub)recent mud volcanic eruptions, but these sites remain unexplored (Khara-Zire, Garasu, Gil and Sangi-Mughan). To the best

of our knowledge, only Baldermann et al. [16] have reported on the mineralogical, textural and chemical compositions of fluid and mud ejects from Zenbil island, which is located in the coastal water of the Caspian Sea. The authors compare the mud mineralogy from Zenbil with onshore material collected from the mud volcano system at Bahar, concluding that the mud has a homogenous composition at the two localities, thus originating from the same shallow-seated mud chamber. In this study, the mineralogical and geochemical compositions of expelled mud, sedimentary rocks and seep water from five prominent mud volcanoes were studied for the first time. The origin of the fluids and the provenance of the mud are discussed.

## 2. Materials and Methods

### 2.1. Materials

The study sites, Hamamdagh and Bendovan, are located in the Salyan region, ~60 km and ~80 km southwest of Baku (Azerbaijan), respectively, and are similar to the onshore mud volcano system at Bahar (Figure 1b, locations 1–2). The sites Khara-Zire, Garasu and Sangi-Mughan are located in the Caspian Sea, ~40 km, ~60 km and ~70 km southwest of Baku, respectively, and are similar to the offshore mud volcano system at Zenbil (Figure 1b, locations 3–5). Thus, the new mud volcanoes studied here are located in a southern direction from the mud volcanoes of Zenbil and Bahar.

Solid and fluid samples were taken from all mud volcanoes during a field campaign carried out in October 2021 (Figure 2). Mud suspensions were collected from gryphon-type emissions within the central mud crater at each study site. The fresh mud suspensions appear as fine-grained sediments of a grey color dispersed in a liquid that is dominated either by water or a water–oil mixture, henceforth called liquid mud and oily mud, respectively (cf. Figure 2a,b). The oily mud contains liquid hydrocarbons and has an oily appearance and a strong smell of hydrogen sulfide. Some mud emissions are mixed with $CH_4$ gas (indicated by bubbling); however, the gas phase was not sampled in this study. In total, 33 solid samples were taken from the different study sites, which include liquid mud suspensions (6 samples, Figure 2a,d,f), oily mud suspensions (5 samples, Figure 2b), as well as 3 types of partly consolidated or fully consolidated sedimentary rocks, light grey to dark grey in color. The latter appear as (i) 'brecciated' mud crusts with desiccation cracks (4 samples, Figures 2e and 3a), (ii) claystones (9 samples, Figures 2c and 3b) and (iii) sandstones (6 samples, Figure 3c). All sample types occasionally contain euhedral pyrite crystals ranging from millimeters to centimeters in size, which were isolated via hand picking (3 samples).

Solid and fluid samples were prepared in the laboratories of the Institute of Geology and Geophysics, Ministry of Science and Education of the Republic of Azerbaijan, and the Institute of Applied Geosciences, Graz University of Technology. The fluids were filtered through 0.45 μm cellulose acetate membrane filters (Sartorius, Vienna, Austria) and separated in three aliquots for subsequent measurements of (i) temperature, alkalinity, pH and electric conductivity (EC), (ii) major cations and anions and (iii) dissolved trace elements. For the trace elemental analysis, the fluid samples were acidified to a 2% nitric acid ($HNO_3$) matrix using $HNO_3$ of suprapure grade (Roth, ROTIPURAN®, Karlsruhe, Germany). The solids were dried at 40 °C and subsequently ground using an agate mortar and pestle.

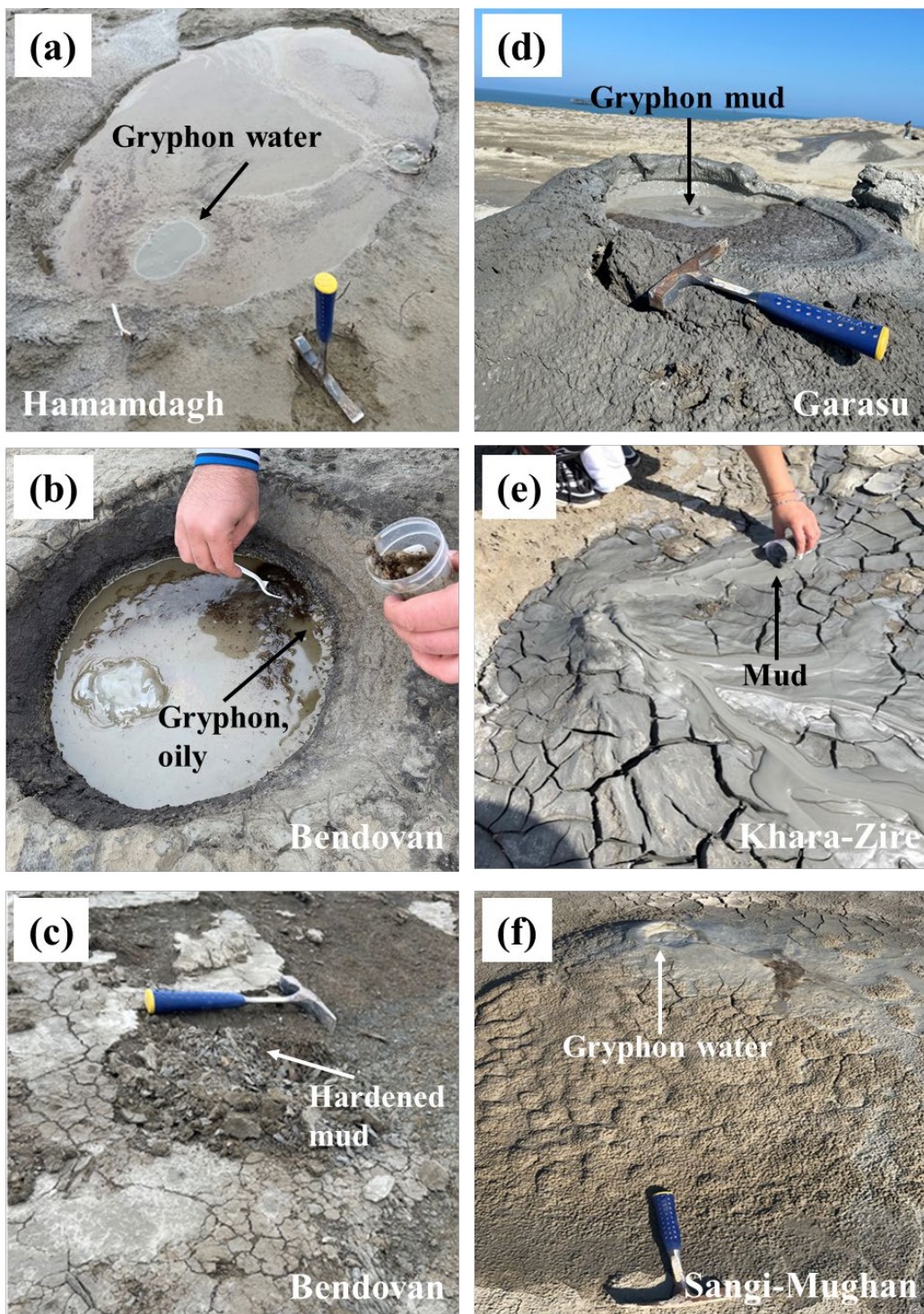

**Figure 2.** Photographs showing field impressions and specific sampling sites at the mud volcanoes in (**a**) Hamamdagh, (**b**,**c**) Bendovan, (**d**) Garasu, (**e**) Khara-Zire and (**f**) Sangi-Mughan, located in the Baku area and on islands within the Caspian Sea, Azerbaijan.

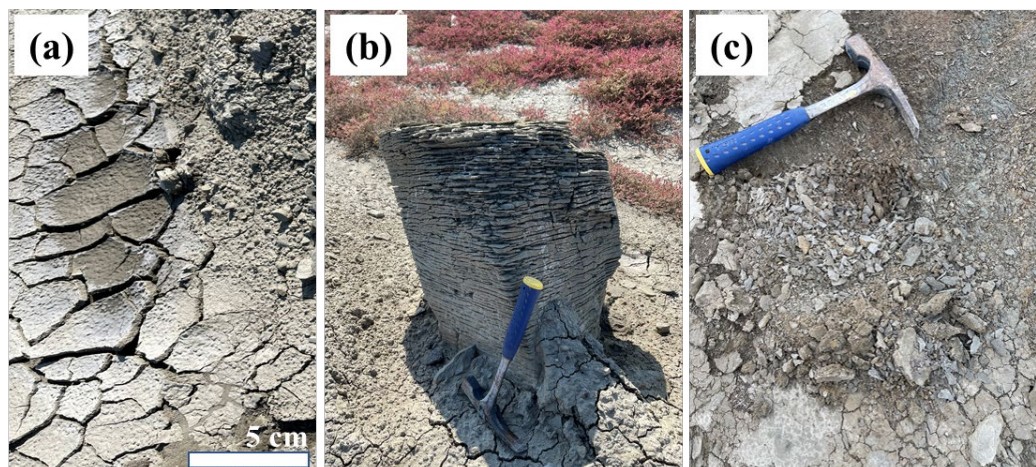

**Figure 3.** Photographs showing mud and sedimentary rock ejects from the mud volcano at Khara-Zire, offshore in the Caspian Sea, Azerbaijan. (**a**) Brecciated mud with desiccation cracks. (**b**) Claystone. (**c**) Sandstone fragments.

### 2.2. Analytical Methods

### 2.2.1. Fluid Phase Characterization

A SenTix 41 glass electrode connected to a WTW Multi 350i pH meter was used to determine the pH and temperature of the extracted fluids. The pH electrode was calibrated using NIST standard buffer solutions at pH 7.00 and 10.00 at 25 °C. The analytical error was ±0.03 pH units. For the determination of the EC values, a WTW LF 330 m connected to a WTW Multi 350i probe was used (analytical error: <3%; WTW GmbH, Weilheim, Germany). Total alkalinity was determined via the automated titration of a 0.1 M hydrochloric acid (HCl) solution using a Schott TitroLine alpha plus titrator (Xylem Analytics Germany Sales GmbH & Co. KG, Weilheim, Germany) with an analytical accuracy of ±2% [29].

A Dionex ICS-3000 ion chromatograph (Thermo Fisher Scientific Inc., Waltham, MA, USA) was used to analyze the concentrations of the major cations (sodium, $Na^+$, potassium, $K^+$, calcium, $Ca^{2+}$ and magnesium, $Mg^{2+}$) and anions (chloride, $Cl^-$, sulphate, $SO_4^{2-}$ and bromide, $Br^-$) in filtered, non-acidified samples with an analytical error of <3% [30]. The total dissolved trace element concentrations (silver, Ag; aluminum, Al; As; boron, B; Ba; cadmium, Cd; cobalt, Co; chromium, Cr; Cu; iron, Fe; lithium, Li; manganese, Mn; nickel, Ni; phosphorous, P; lead, Pb; silicon, Si; Sr; and Zn) were analyzed in acidified aliquots using a PerkinElmer Optima 8300 DV optical emission spectrometer with inductively coupled plasma (PerkinElmer, Waltham, MA, USA). The analytical precision was better than ±3% (2σ, 3 replicates), as determined by replicate analyses of NIST 1640a, as well as in-house and SPS-SW2 Batch 130 (Merck KGaA, Darmstadt, Germany) standards [31].

The PHREEQC software (version 2.18.00) with its llnl.dat database was used to calculate the species distribution, activities, ionic strength, ionic charge balances and saturation indices (SI) of the fluids with respect to relevant mineral phases, such as feldspars, carbonates and clay minerals [32].

Fluid generation temperatures were calculated using common chemical geo-thermometers (Na-K, K-Mg, and $SiO_2$), which were originally designed for and applied to low- to high-salinity pore fluids generated in oil and gas basins [33–35]:

$$T_{Na/K} \ (^\circ C) = 1217/(1.48 + \log(Na/K)) - 273.15$$

$$T_{K/Mg} \ (^\circ C) = 4410/(14.0 - \log(K^2/Mg) - 273.15$$

$$T_{SiO2} \ (^\circ C) = 1522/(5.75 - \log(SiO_2)) - 273.15$$

A discussion of the applicability and limitations of these geo-thermometers is provided elsewhere [36].

### 2.2.2. Solid Phase Characterization

The quantitative mineralogical composition of the solids was determined using powder X-ray diffraction (P-XRD). The top loading method was applied to prepare the powdered materials in standard XRD sample holders [37], which were examined using a PANalytical X'Pert PRO diffractometer in a range from 4 to 85° 2θ with a step of 0.008° 2θ and a count time of 40 s per step. This device is equipped with a Scientific X'Celerator detector and a Co-Kα X-ray radiation source operating at 40 kV and 40 mA. Mineral identification was carried out using the PANalytical X'Pert Highscore Plus software (version 2.2e) and a pdf-4 database, based on the following criteria: quartz (peaks at 4.25 Å and 3.34 Å), calcite (peaks at 3.85 Å and 3.03 Å), dolomite (peak at ~2.9 Å for dolomite stoichiometry; intensity ratio of the $d_{(105)}/d_{(110)}$ peaks for dolomite ordering degree), albite (peak at 3.19 Å), orthoclase (peak at 3.24 Å), pyrite (peaks at 2.71 Å and 1.63 ÅÅ), chlorite (peaks at ~14.2 Å, 4.71 Å and 1.54 Å), kaolinite (peaks at 7.15 Å, 3.57 Å and 1.49 Å), illite (peaks at ~10 Å, 4.99 Å and 1.50 Å) and smectite (broad peaks at ~13.5 Å and 1.50 Å). Randomly oriented mounts (<2 μm size fraction) were prepared and X-rayed as indicated before, each in air-dried, glycolated and heated (550 °C for 1 h) states, to exclude mixed-layered phases in the samples. Subsequently, mineral quantification was carried out via Rietveld refinement of P-XRD patterns using the PANalytical X'Pert Highscore Plus software (version 2.2e) and its implemented ICSD database with an analytical error of <3 wt.%, judged by comparing with chemical data and interlaboratory, cross-method calibration [38].

The chemical composition of the solids was analyzed using an Epsilon 4 benchtop energy-dispersive X-ray fluorescence (XRF) spectrometer. Approximately 1.0 g of sample was heated to 950 °C for 1 h to remove volatiles. Then, the loss on ignition (LOI) was determined via standard gravimetric analysis. Glass tablets were produced via the fusion of 6.0 g of $Li_2B_4O_7$ (fluid agent) and 0.6 g of sample at ~1200 °C in a PANalytical Perl X 3 bead preparation system (Malvern Panalytical, Malvern, United Kingdom). To avoid volatilization and loss of sulfur, the fusion time was reduced to ~3 min. The glass tablets were measured together with a set of USGS standards. For the major elements (expressed as oxides), the analytical error was <0.5 wt.% [39].

The Chemical Index of Alteration (CIA) was calculated from these chemical data, according to the expression $(Al_2O_3/(Al_2O_3 + Na_2O + K_2O + CaO^*))\cdot100$ [40], to quantitatively determine the weathering intensity of the solids ejected from the mud volcanoes. Therefore, CaO related to carbonate minerals was subtracted from the total CaO content to obtain the CaO* content of the silicate fraction [41]. The chemical compositions were also plotted in the A–CN–K ($Al_2O_3 - CaO^* + Na_2O - K_2O$) diagram to identify alteration paths within the mud volcanic ejects [40].

## 3. Results and Discussion

### 3.1. Chemical Composition of the Expelled Fluids

The chemical compositions of the fluids expelled from the gryphon-type emissions are reported in Table 1. The fluids extracted from the offshore mud volcanoes in Hamamdagh and Bendovan have pH values of 8.32 and 8.55, which is well within the range of published values since the pH of ~130 mud volcanoes from Azerbaijan varies between 6.51 and 8.84 [16,26,28]. These values are slightly lower compared to the onshore mud volcanoes at Garasu, Khara-Zire and Sangi-Mughan, which exhibit pH values between 8.58 and 9.10. The EC values of all fluids range from 23.3 and 40.8 mS/cm without a systematic geographic trend. The plot of the water compositions in the Piper diagram reveals a Na–Cl-type composition for all fluids (Figure 4), as well as a large overlap with the water chemistries reported for Bahar and Zenbil [4,16]. This suggests that a large fraction of the fluids is sourced from concentrated Caspian seawater due to evaporation.

**Table 1.** Composition of the fluids expelled from the mud volcanoes in Hamamdagh, Bendovan, Garasu, Khara-Zire and Sangi-Mughan, Azerbaijan. Alkalinity is expressed as $HCO_3$. Reported EC values correspond to total dissolved solids (TDS) contents of 12.1, 20.4, 11.7, 13.4 and 12.8 g/kg of fluids, respectively. Ion charge balance errors are below 2%. The composition of Caspian seawater is included for comparison (n.a.—not analyzed).

| Sample ID | SpC (mS/cm) | T (°C) | pH (−) | Na (mg/L) | K (mg/L) | Mg (mg/L) | Ca (mg/L) | Cl (mg/L) |
|---|---|---|---|---|---|---|---|---|
| Garasu | 24.1 | 21.4 | 8.75 | 8343 | 18 | 71 | 51 | 12,702 |
| Sangi-Mughan | 40.8 | 22.0 | 8.58 | 13,919 | 35 | 109 | 9 | 19,756 |
| Hamamdag | 23.3 | 21.6 | 8.32 | 8721 | 20 | 94 | 57 | 12,552 |
| Byandovan | 26.8 | 21.5 | 8.55 | 7010 | 17 | 82 | 94 | 10,192 |
| Khara-Zire | 25.5 | 21.8 | 9.10 | 7108 | 16 | 12 | 9 | 9267 |

| Sample ID | Br (mg/L) | SO$_4$ (mg/L) | HCO$_3$ (mg/L) | B (mg/L) | Al (µg/L) | Ba (µg/L) | Cu (µg/L) | Fe (µg/L) |
|---|---|---|---|---|---|---|---|---|
| Garasu | 48 | 33 | 1441 | 17 | 146 | 1063 | 84 | <1 |
| Sangi-Mughan | 84 | 842 | 2868 | 13 | 453 | 202 | 117 | <1 |
| Hamamdag | 78 | 390 | 1546 | 11 | 155 | 709 | 131 | 83 |
| Byandovan | 54 | 285 | 1843 | 19 | 190 | 10,808 | 117 | <1 |
| Khara-Zire | 47 | 9 | 3190 | 32 | 61 | 6372 | 59 | <1 |

| Sample ID | Li (µg/L) | Si (µg/L) | Sr (µg/L) | Zn (µg/L) | As (µg/L) | Na–K (°C) | K–Mg (°C) | SiO$_2$ (°C) |
|---|---|---|---|---|---|---|---|---|
| Garasu | 420 | 2151 | 8585 | 642 | 796 | 20 | 57 | 26 |
| Sangi-Mughan | 2271 | 1893 | 3085 | 535 | 658 | 25 | 67 | 23 |
| Hamamdag | 1295 | 3748 | 19,104 | 1023 | 1141 | 22 | 57 | 41 |
| Byandovan | 940 | 3422 | 26,338 | 1023 | 1420 | 24 | 55 | 38 |
| Khara-Zire | 1149 | 3269 | 4930 | 494 | 618 | 22 | 75 | 37 |

Accordingly, the fluids are dominated by $Na^+$ and $Cl^-$ ions, but there is also a considerable range in concentration of the other cations, such as $Mg^{2+}$, $K^+$ and $Ca^{2+}$, and anions, such as hydrogen carbonate $HCO_3^-$, $SO_4^{2-}$, $Br^-$ and $B(OH)_3$ (Table 1), which overlap with compositions reported for other mud volcanoes from Azerbaijan [3,4,16,26,28]. The concentrations of $K^+$, $Mg^{2+}$, $Ca^{2+}$ and $SO_4^{2-}$ are 3 to 370 times lower compared to Caspian seawater, whereas $Na^+$, $Cl^-$, $B(OH)_3$, $Br^-$ and $HCO_3^-$ are enriched by between 2 and 15 times (Table 1). This is due to site-specific mineral dissolution/precipitation reactions, such as smectite illitization, the thermal maturation of organic matter and $CH_4$ production, dissolution of evaporites exposed in the Maykop Suite shales, redox reactions and weathering processes [11,16,42–47].

All fluids contain a variable, but generally high concentration of dissolved metal ions, such as Al, As, Ba, Cu, Li, Si, Sr and Zn (Table 1), which are similar to mud volcano compositions reported in previous studies [4,11,16,26]. Some of these components are up to several hundred times enriched relative to Caspian seawater. By contrast, the concentration of dissolved Fe is extremely low (Table 1), which is due to the precipitation of iron sulfides, corroborating findings for euhedral pyrite crystals in the mud matrix. Furthermore, this observation is consistent with occurrences of cubic pyrite found in many mud volcanoes across Azerbaijan [48]. These are formed under reducing conditions, as indicated by Eh values of ~30 mud volcanoes in Azerbaijan ranging from −20 to −295 mV [26].

The fluids expelled from the studied mud volcanoes are supersaturated with respect to carbonate minerals, such as aragonite, calcite and dolomite, Fe- and Al-(hydr)oxides (e.g., goethite and gibbsite) and clay minerals, such as Mg-chlorite, illite, Na-montmorillonite and kaolinite (Table 2). Albite, orthoclase and barite are close to saturation or supersaturated with respect to the fluids, whereas anorthite, quartz and evaporite minerals, such as gypsum and halite, are undersaturated (Table 2). No systematic trends in the mineral saturation indices are seen among the different localities.

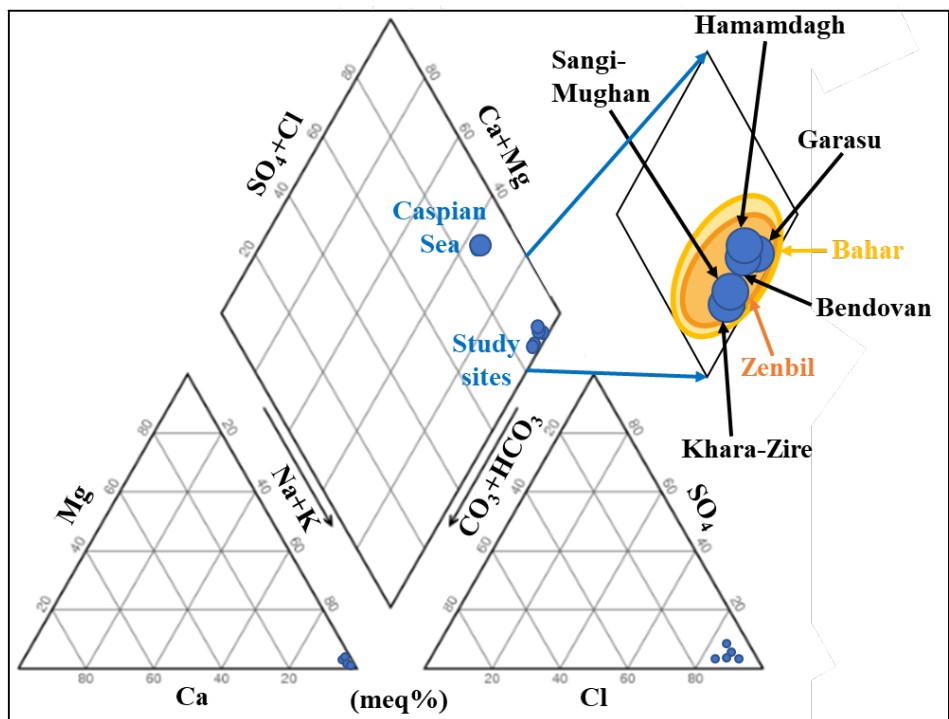

**Figure 4.** Piper trilinear diagram used for the classification of seep waters collected from the mud volcanoes in Hamamdagh, Bendovan, Garasu, Khara-Zire and Sangi-Mughan, Azerbaijan. Caspian seawater and compositions of fluids ejected in Bahar and Zenbil are included for comparison.

**Table 2.** Saturation indices of various mineral phases with respect to the fluids expelled from the mud volcanoes in Hamamdagh, Bendovan, Garasu, Khara-Zire and Sangi-Mughan, Azerbaijan. Mineral abbreviations: Ab—albite, An—anorthite, Arg—aragonite, Brt—barite, Cal—calcite, Chl—chlorite, Dol—dolomite, Gbs—gibbsite, Gth—goethite, Gp—gypsum, Hl—halite, Ilt—illite, Kln—kaolinite, Mnt—montmorillonite, Qz, quartz.

| Sample ID | Arg (−) | Cal (−) | Dol (−) | Gth (−) | Gbs (−) | Chl (−) | Ilt (−) | Na-Mnt (−) |
|---|---|---|---|---|---|---|---|---|
| Garasu | 1.3 | 1.5 | 4.6 | 4.7 | 1.0 | 8.3 | 1.5 | 0.9 |
| Sangi-Mughan | 0.6 | 0.7 | 4.0 | 4.7 | 1.6 | 8.2 | 2.7 | 1.7 |
| Hamamdag | 1.0 | 1.2 | 4.0 | 5.7 | 1.5 | 6.8 | 3.7 | 2.9 |
| Byandovan | 1.5 | 1.7 | 4.8 | 4.7 | 1.3 | 8.1 | 3.1 | 2.4 |
| Khara-Zire | 1.0 | 1.1 | 3.9 | 4.6 | 0.5 | 6.6 | 0.5 | 0.2 |
| Sample ID | Kln (−) | Ab (−) | K-Fsp (−) | Brt (−) | An (−) | Qz (−) | Gp (−) | Hl (−) |
| Garasu | 1.3 | 0.1 | 0.3 | 0.0 | −4.5 | −0.7 | −3.4 | −2.8 |
| Sangi-Mughan | 2.5 | 0.6 | 0.8 | 0.5 | −4.5 | −0.7 | −2.9 | −2.4 |
| Hamamdag | 3.2 | 1.4 | 1.5 | 0.8 | −3.4 | −0.3 | −2.2 | −2.8 |
| Byandovan | 2.6 | 1.1 | 1.3 | 2.0 | −3.4 | −0.4 | −2.1 | −3.0 |
| Khara-Zire | 0.3 | 0.0 | 0.1 | 0.2 | −5.7 | −0.6 | −4.7 | −3.0 |

### 3.2. Fluid Origin

The $Na^+$ and $Cl^-$ concentrations in all sampled fluids are up to 6 times higher than Caspian seawater, but the molar Na/Cl ratio of $1.0 \pm 0.1$ suggests that the fluids are mainly derived from concentrated Caspian seawater (Figure 4). However, the plot of the Cl–B and Cl–Br relations indicates that the fluids have a mixed origin, which can be described using a two-component mixing model (Figure 5). Type–1 waters have salinities resembling slightly evaporated Caspian seawater, as well as B and Br systematics that are typical of 'low-mineralized' pore fluids, which are sourced from Maykop Suite shales and the Productive

Series strata [4,22,49]. Type–2 waters have a higher degree of mineralization and elevated B and Br concentrations originating from concentrated ancient (Eastern Paratethys-derived) sedimentary pore fluids, which are interpreted as 'high-mineralized' oilfield brines [4,50]. Thus, all fluids are generated via the mixing of modern Caspian seawater and ancient brines sourced from shallow, but not meteoric, to deep reservoir levels. Mazzini et al. [11] and Aliyev et al. [21] argue that the fluids are generated at different depths, including (i) shallow mud chambers within the Productive Series strata (~1 to 5.5 km depth), (ii) moderately deep mud chambers within the Miocene Maykop Suite shales (~5.5 to 9 km depth) and (iii) very deep mud chambers within Eocene-aged strata (up to ~12 km depth). The same conclusions have previously been drawn for the mud and fluid ejects from the mud volcano systems at Bahar and Zenbil [16].

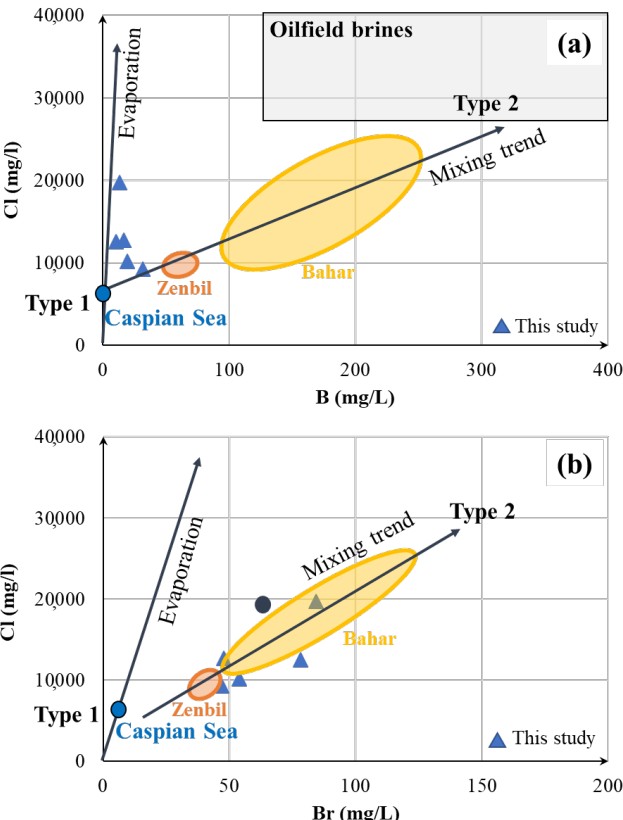

**Figure 5.** Cross-plots of (**a**) Cl vs. B and (**b**) Cl vs. Br concentrations measured in seep waters. The Cl–B–Br relations of the fluids from the mud volcanoes at Hamamdagh, Bendovan, Garasu, Khara-Zire and Sangi-Mughan (Azerbaijan) reveal a mixed origin, similar to the mud volcanoes at Bahar and Zenbil [16]: Type–1 water = concentrated Caspian seawater; Type–2 water = oilfield brines or ancient sedimentary (highly saline) pore fluids [50]. The ideal evaporation line for Caspian seawater is included for comparison.

The generation temperatures of the fluids range from 20 to 41 °C, with an average of $28 \pm 8$ °C, as constrained by Na–K and $SiO_2$ geo-thermometers (Table 1). The applicability of such thermodynamic calculations requires mineral–fluid equilibria, which are barely developed in dynamic mud volcano systems. However, the fluids expelled at Khara-Zire are close to equilibrium with respect to illite and smectite (Table 2), and the calculated mud generation temperatures are comfortably in the range of those calculated for the other localities, suggesting that the obtained generation temperatures of the fluids are correct and realistic within the uncertainty of the geo-thermometers [36]. No systematic trends are seen among the different localities. The K–Mg geo-thermometer yields a higher generation temperature of 55 to 75 °C, averaging $62 \pm 9$ °C (Table 1), which is too high for the study

sites investigated here. We attribute this to the removal of $Mg^{2+}$ ions from the fluids via the precipitation of dolomite and smectite minerals (see below), thus limiting the reliability of the geo-thermometer. Considering a temperature range from 20 to 41 °C and a geothermal gradient of 10–18 °C/km (average: 15 °C/km) for the South Caspian Sea [5], the formation depth of the fluids varies in the range from ~1.1 to 4.1 km for all localities. Such a shallow origin of the fluids is consistent with previous estimates of mud volcanoes in Bahar and Zenbil, which originate from depths from ~1.8 to 4.2 km [16].

### 3.3. Mineralogical Composition of Solid Ejects

The mineralogy of the mud samples is identical regarding the liquid, oily and brecciated forms and no geographic differences are observable among the individual localities (Figure 6). The mud is mainly composed of quartz, calcite, calcian dolomite with ~55 mol% $CaCO_3$ and a cation ordering ratio of 0.3–0.5, albite, orthoclase, pyrite and clay minerals, such as illite, chlorite, Na-smectite and kaolinite (Table 3). The Ca excess and low ordering degree of the largely rhombohedral dolomite suggest an authigenic origin of this phase [51], corroborating fluid chemical data (Table 2). Pyrite appears as idiomorphic cubes ranging from millimeters to centimeters in size, also indicating the authigenic origin of this phase [52]. The clay mineral assemblage most likely has a mixed origin and is composed of detrital, platy illite and chlorite, in addition to dominantly authigenic, veil-like to filmy smectite and platelet kaolinite; the latter phases likely represent weathering products of feldspar (Table 2). Sub-rounded quartz, feldspar and likely a fraction of the clay minerals and carbonates are also of detrital origin, being originated from the different sedimentological units the mud is passing through during its ascent. The above described particle forms resemble the prior mineral shapes described by Baldermann et al. [16], which are judged via an SEM (with EDS) analysis during the initial sample inspection, and thus are not shown.

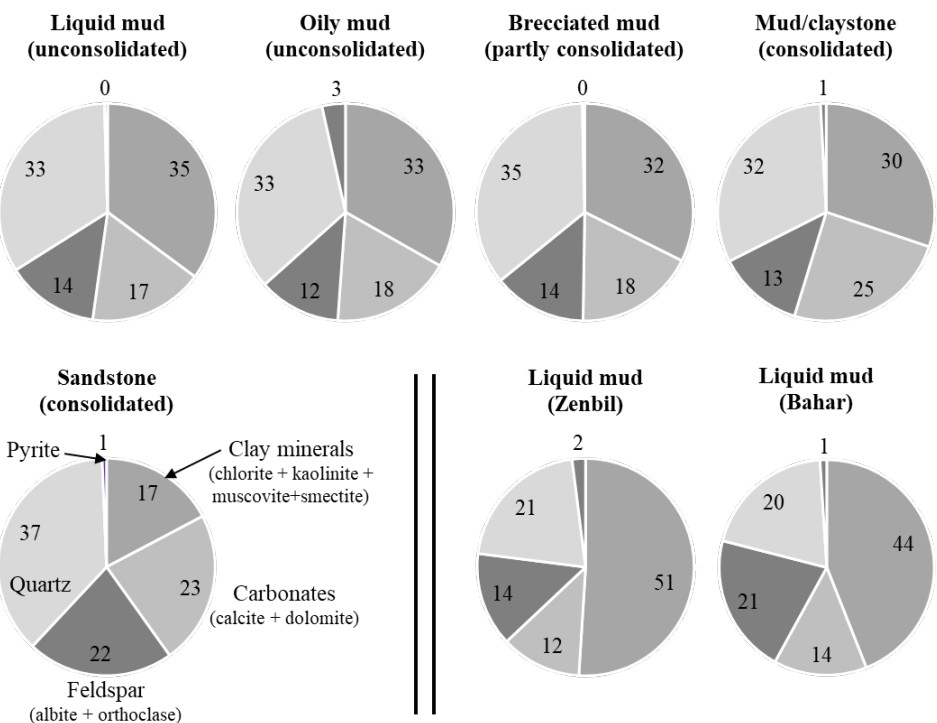

**Figure 6.** Comparison of averaged mineralogical compositions obtained for liquid mud (n = 6), oily mud (n = 5), brecciated mud (n = 4), mud/claystones (n = 9) and sandstones (n = 5) collected from the mud volcanoes in Hamamdagh, Bendovan, Garasu, Khara-Zire and Sangi-Mughan, Azerbaijan (cf. Figures 2 and 3 for field impressions). Mud compositions from Bahar and Zenbil are included for comparison (data are from [16]).

**Table 3.** Mineralogical composition of the mud and sedimentary rocks expelled from the mud volcanoes in Hamamdagh, Bendovan, Garasu, Khara-Zire and Sangi-Mughan, Azerbaijan. The average mineralogical composition of the liquid mud ejects from Bahar (n = 18) and Zenbil (n = 13) is included for comparison (data are from [16]).

| Sampling Site | Sample Type | Quartz (wt.%) | Calcite (wt.%) | Dolomite (wt.%) | Albite (wt.%) | Orthoclase (wt.%) |
|---|---|---|---|---|---|---|
| Hamamdagh | Liquid mud | 32 | 13 | 8 | 8 | 2 |
| Hamamdagh | Liquid mud | 31 | 14 | 1 | 12 | 3 |
| Khara-Zire | Liquid mud | 40 | 17 | 3 | 11 | 2 |
| Garasu | Liquid mud | 28 | 14 | 1 | 12 | 2 |
| Sangi-Mughan | Liquid mud | 37 | 14 | 1 | 13 | 2 |
| Sangi-Mughan | Liquid mud | 33 | 14 | 2 | 15 | 2 |
| Hamamdagh | Oily mud | 30 | 16 | 2 | 11 | 2 |
| Bendovan | Oily mud | 36 | 16 | 1 | 7 | 3 |
| Khara-Zire | Oily mud | 39 | 14 | 1 | 11 | 1 |
| Garasu | Oily mud | 32 | 16 | 1 | 11 | 3 |
| Sangi-Mughan | Oily mud | 29 | 22 | 1 | 11 | 2 |
| Hamamdagh | Brecciated mud | 35 | 16 | 5 | 13 | 3 |
| Bendovan | Brecciated mud | 29 | 15 | 2 | 9 | 2 |
| Khara-Zire | Brecciated mud | 42 | 14 | 2 | 13 | 1 |
| Garasu | Brecciated mud | 36 | 15 | 3 | 12 | 2 |
| Hamamdagh | Mud/claystone | 29 | 28 | 2 | 14 | 2 |
| Hamamdagh | Mud/claystone | 22 | 9 | 35 | 7 | 2 |
| Bendovan | Mud/claystone | 30 | 21 | 1 | 10 | 2 |
| Bendovan | Mud/claystone | 36 | 21 | 1 | 9 | 2 |
| Khara-Zire | Mud/claystone | 38 | 12 | 7 | 13 | 1 |
| Garasu | Mud/claystone | 32 | 15 | 2 | 12 | 2 |
| Garasu | Mud/claystone | 32 | 14 | 7 | 12 | 2 |
| Sangi-Mughan | Mud/claystone | 33 | 17 | 8 | 11 | 1 |
| Sangi-Mughan | Mud/claystone | 32 | 14 | 8 | 10 | 3 |
| Bendovan | Sandstone | 25 | 25 | 1 | 21 | 3 |
| Bendovan | Sandstone | 31 | 18 | 2 | 21 | 4 |
| Khara-Zire | Sandstone | 55 | 11 | 12 | 14 | 2 |
| Garasu | Sandstone | 39 | 17 | 6 | 20 | 3 |
| Sangi-Mughan | Sandstone | 37 | 18 | 4 | 19 | 3 |
| Zenbil | Liquid mud | 21 | 10 | 2 | 14 | 1 |
| Bahar | Liquid mud | 20 | 13 | 1 | 21 | 0 |

| Sampling Site | Sample Type | Pyrite (wt.%) | Chlorite (wt.%) | Kaolinite (wt.%) | Illite (wt.%) | Smectite (wt.%) |
|---|---|---|---|---|---|---|
| Hamamdagh | Liquid mud | 1 | 19 | 1 | 8 | 9 |
| Hamamdagh | Liquid mud | 0 | 11 | 2 | 17 | 11 |
| Khara-Zire | Liquid mud | 1 | 8 | 2 | 13 | 4 |
| Garasu | Liquid mud | 0 | 8 | 2 | 24 | 9 |
| Sangi-Mughan | Liquid mud | 1 | 11 | 2 | 16 | 4 |
| Sangi-Mughan | Liquid mud | 0 | 8 | 2 | 17 | 5 |
| Hamamdagh | Oily mud | 1 | 11 | 1 | 22 | 3 |
| Bendovan | Oily mud | 12 | 6 | 2 | 9 | 9 |
| Khara-Zire | Oily mud | 1 | 8 | 2 | 17 | 5 |
| Garasu | Oily mud | 1 | 11 | 2 | 18 | 6 |
| Sangi-Mughan | Oily mud | 3 | 8 | 2 | 16 | 6 |
| Hamamdagh | Brecciated mud | 0 | 8 | 1 | 16 | 3 |
| Bendovan | Brecciated mud | 0 | 13 | 1 | 10 | 19 |
| Khara-Zire | Brecciated mud | 1 | 9 | 1 | 10 | 6 |
| Garasu | Brecciated mud | 0 | 10 | 1 | 14 | 7 |

**Table 3.** *Cont.*

| | | | | | | |
|---|---|---|---|---|---|---|
| Hamamdagh | Mud/claystone | 0 | 13 | 2 | 6 | 4 |
| Hamamdagh | Mud/claystone | 0 | 12 | 2 | 9 | 2 |
| Bendovan | Mud/claystone | 0 | 11 | 2 | 17 | 6 |
| Bendovan | Mud/claystone | 0 | 8 | 2 | 14 | 7 |
| Khara-Zire | Mud/claystone | 2 | 4 | 1 | 16 | 5 |
| Garasu | Mud/claystone | 1 | 10 | 2 | 20 | 5 |
| Garasu | Mud/claystone | 1 | 11 | 1 | 15 | 5 |
| Sangi-Mughan | Mud/claystone | 1 | 9 | 1 | 14 | 5 |
| Sangi-Mughan | Mud/claystone | 1 | 9 | 2 | 17 | 5 |
| Bendovan | Sandstone | 1 | 8 | 2 | 5 | 11 |
| Bendovan | Sandstone | 1 | 5 | 1 | 7 | 9 |
| Khara-Zire | Sandstone | 0 | 1 | 1 | 5 | 0 |
| Garasu | Sandstone | 1 | 7 | 1 | 3 | 2 |
| Sangi-Mughan | Sandstone | 1 | 5 | 1 | 7 | 6 |
| Zenbil | Liquid mud | 2 | 3 | 8 | 26 | 13 |
| Bahar | Liquid mud | 1 | 4 | 6 | 22 | 12 |

The mud/claystones have slightly lower contents of clay minerals and quartz, a similar feldspar content and a higher carbonate content compared to the mud. The sandstones have the highest contents of quartz and feldspar and moderate carbonate and the lowest clay mineral contents (Figure 6). Compared to the mud compositions reported for Bahar and Zenbil [16], the mud volcanoes investigated in this study have a slightly lower clay mineral content and contain higher amounts of quartz and carbonates (Figure 6).

*3.4. Geochemical Composition of Solid Ejects*

The geochemical composition of the mud and sedimentary rocks ejected from the mud volcanoes in Hamamdagh, Bendovan, Garasu, Khara-Zire and Sangi-Mughan is summarized in Table 4. The major and minor elemental compositions obtained for all deposit types show minor sample/rock-specific variations, corroborating mineralogical results (Figure 6 and Table 3). Accordingly, the liquid, oily and brecciated mud ejects from all localities have the same chemical composition within analytical uncertainty, averaging $47 \pm 5$ wt.% $SiO_2$, $13 \pm 1$ wt.% $Al_2O_3$, $16 \pm 3$ wt.% LOI, $8 \pm 1$ wt.% CaO, $7 \pm 2$ wt.% $Fe_2O_3$, $3 \pm 1$ wt.% $Na_2O$, $3 \pm 1$ wt.% MgO, $2 \pm 1$ wt.% $K_2O$ and $1 \pm 1$ wt.% $SO_3$, with minor $TiO_2$, MnO and $P_2O_5$ (<1 wt.%). This composition reflects the abundance of quartz, feldspar, carbonates, clay minerals, pyrite and Ti-bearing phases (most likely rutile) in mud, as previously indicated (Table 3). The mud/claystones exhibit a larger scatter in their CaO, MgO and LOI contents compared to the mud, reflecting the higher proportion of carbonate minerals in this rock type, whereas the sandstones have higher contents of $SiO_2$ and CaO and lower contents of LOI, $Al_2O_3$ and $Fe_2O_3$, which is consistent with increased fractions of quartz and feldspar and reduced amounts of clay minerals in this rock type (Figure 6). As for all eject types, the weak but positive correlation of rubidium (Rb) and vanadium (V) with $Al_2O_3$ ($R^2$ = 0.3 and 0.4), yttrium (Y; $R^2$ = 0.5 and 0.5), $TiO_2$ ($R^2$ = 0.3 and 0.5), Ni ($R^2$ = 0.3 and 0.4) and $SO_3$ ($R^2$ = 0.3 and 0.4) suggests an association with clay minerals, heavy minerals and sulfide minerals, respectively [41]. Barium, Co, Cr, Cu, gallium (Ga), molybdenum (Mo), niobium (Nb), Pb, Sr and Zn are inconspicuous due to either a lack of a correlation with major elements or low concentrations in the samples. However, high Ni/Co ratios (mostly above 5) and low Cu/Zn ratios (mostly below 1) indicate that dysoxic to anoxic conditions prevail within the mud chamber [53], which is consistent with authigenic pyrite found in the mud matrix [48].

**Table 4.** Geochemical composition of the mud and sedimentary rocks expelled from the mud volcanoes in Hamamdagh, Bendovan, Garasu, Khara-Zire and Sangi-Mughan, Azerbaijan. The average composition of the liquid mud ejects from Bahar and Zenbil is included for comparison (data are from [16]). n.a.—not analyzed. The concentrations of Ga (<20 ppm), Mo (<10 ppm) and Nb (<20 ppm) are always below the detection limit of the XRF analysis.

| Sampling Site | Sample Type | $Na_2O$ (wt.%) | MgO (wt.%) | $Al_2O_3$ (wt.%) | $SiO_2$ (wt.%) | $P_2O_5$ (wt.%) | $SO_3$ (wt.%) | $K_2O$ (wt.%) | CaO (wt.%) |
|---|---|---|---|---|---|---|---|---|---|
| Hamamdagh | Liquid mud | 3.0 | 3.2 | 13.7 | 49.8 | 0.1 | 0.4 | 2.3 | 6.2 |
| Hamamdagh | Liquid mud | 2.7 | 3.0 | 13.0 | 44.8 | 0.1 | 1.0 | 2.3 | 8.1 |
| Khara-Zire | Liquid mud | 2.6 | 3.2 | 11.7 | 48.9 | 0.1 | 0.3 | 1.9 | 10.7 |
| Garasu | Liquid mud | 2.9 | 2.7 | 13.0 | 48.9 | 0.1 | 0.3 | 1.9 | 8.0 |
| Sangi-Mughan | Liquid mud | 3.2 | 2.6 | 12.5 | 47.6 | 0.1 | 0.8 | 1.9 | 7.0 |
| Sangi-Mughan | Liquid mud | 3.3 | 2.7 | 13.3 | 48.9 | 0.1 | 0.3 | 2.0 | 6.7 |
| Hamamdagh | Oily mud | 3.3 | 2.1 | 13.2 | 49.9 | 0.1 | 0.9 | 2.0 | 6.7 |
| Bendovan | Oily mud | 2.6 | 2.5 | 8.8 | 32.3 | 0.4 | 4.1 | 1.4 | 9.7 |
| Khara-Zire | Oily mud | 4.0 | 2.3 | 13.3 | 48.5 | 0.1 | 0.9 | 2.0 | 7.1 |
| Garasu | Oily mud | 2.9 | 2.6 | 11.8 | 48.9 | 0.1 | 1.4 | 2.2 | 9.1 |
| Sangi-Mughan | Oily mud | 4.0 | 2.6 | 12.2 | 41.5 | 0.1 | 2.5 | 1.9 | 8.9 |
| Hamamdagh | Brecciated mud | 2.1 | 3.1 | 13.5 | 50.3 | 0.1 | 0.4 | 2.4 | 7.3 |
| Bendovan | Brecciated mud | 2.8 | 4.0 | 13.4 | 43.0 | 0.1 | 0.2 | 2.4 | 9.5 |
| Khara-Zire | Brecciated mud | 2.4 | 2.4 | 13.5 | 50.9 | 0.1 | 1.1 | 2.4 | 7.4 |
| Garasu | Brecciated mud | 2.0 | 2.6 | 13.2 | 50.6 | 0.1 | 0.6 | 2.2 | 8.4 |
| Hamamdagh | Mud/claystone | 3.4 | 3.3 | 12.6 | 47.3 | 0.1 | 0.4 | 1.8 | 8.0 |
| Hamamdagh | Mud/claystone | 1.9 | 4.4 | 12.4 | 42.9 | 0.1 | 0.3 | 2.2 | 9.1 |
| Bendovan | Mud/claystone | 2.6 | 2.6 | 14.7 | 46.0 | 0.1 | 0.2 | 2.3 | 8.5 |
| Bendovan | Mud/claystone | 2.3 | 2.4 | 13.6 | 44.7 | 0.1 | 0.1 | 2.2 | 8.5 |
| Khara-Zire | Mud/claystone | 3.5 | 2.3 | 12.0 | 46.9 | 0.1 | 1.4 | 2.4 | 7.3 |
| Garasu | Mud/claystone | 2.3 | 2.3 | 14.1 | 45.4 | 0.1 | 1.3 | 2.6 | 8.8 |
| Garasu | Mud/claystone | 2.6 | 2.4 | 13.2 | 51.0 | 0.1 | 0.1 | 2.0 | 6.7 |
| Sangi-Mughan | Mud/claystone | 2.5 | 2.3 | 13.7 | 53.9 | 0.1 | 0.4 | 2.0 | 6.0 |
| Sangi-Mughan | Mud/claystone | 1.8 | 1.4 | 12.8 | 63.8 | 0.1 | 0.2 | 2.3 | 0.5 |
| Bendovan | Sandstone | 3.1 | 2.5 | 11.8 | 44.9 | 0.1 | 1.6 | 1.5 | 13.6 |
| Bendovan | Sandstone | 2.9 | 2.0 | 12.3 | 51.2 | 0.1 | 1.3 | 2.1 | 10.0 |
| Khara-Zire | Sandstone | 1.3 | 1.5 | 9.4 | 58.6 | 0.1 | 0.5 | 1.5 | 12.2 |
| Garasu | Sandstone | 3.1 | 2.6 | 12.8 | 55.9 | 0.1 | 0.3 | 1.8 | 7.0 |
| Sangi-Mughan | Sandstone | 3.0 | 2.2 | 11.6 | 53.8 | 0.1 | 0.7 | 1.7 | 11.1 |
| Zenbil | Liquid mud | 2.3 | 2.3 | 14.3 | 52.3 | 0.1 | 1.1 | 1.9 | 6.4 |
| Bahar | Liquid mud | 3.2 | 2.3 | 13.2 | 50.4 | 0.1 | 0.6 | 1.6 | 8.2 |

| Sampling Site | Sample Type | CaO* (wt.%) | $TiO_2$ (wt.%) | MnO (wt.%) | $Fe_2O_3$ (wt.%) | LOI (wt.%) | Ba ppm | Co ppm | Cr ppm |
|---|---|---|---|---|---|---|---|---|---|
| Hamamdagh | Liquid mud | 0.2 | 0.6 | 0.1 | 6.0 | 14.7 | 455 | <20 | 122 |
| Hamamdagh | Liquid mud | 0.1 | 0.7 | 0.1 | 6.4 | 17.7 | 454 | 21 | 112 |
| Khara-Zire | Liquid mud | 0.3 | 0.6 | 0.2 | 5.9 | 13.9 | 442 | <20 | 153 |
| Garasu | Liquid mud | 0.3 | 0.6 | 0.1 | 5.8 | 15.5 | 403 | <20 | 116 |
| Sangi-Mughan | Liquid mud | 0.2 | 0.6 | 0.1 | 5.7 | 18.0 | 386 | <20 | 128 |
| Sangi-Mughan | Liquid mud | 0.4 | 0.6 | 0.1 | 5.8 | 16.2 | 753 | <20 | 108 |
| Hamamdagh | Oily mud | 0.0 | 0.6 | 0.1 | 5.5 | 15.4 | 416 | <20 | 94 |
| Bendovan | Oily mud | 0.3 | 0.6 | 0.1 | 13.2 | 24.2 | 369 | 43 | 118 |
| Khara-Zire | Oily mud | 0.5 | 0.6 | 0.1 | 5.6 | 15.5 | 480 | <20 | 88 |
| Garasu | Oily mud | 0.2 | 0.7 | 0.1 | 6.5 | 13.7 | 400 | 21 | 177 |
| Sangi-Mughan | Oily mud | 0.4 | 0.9 | 0.1 | 7.2 | 18.0 | 484 | 23 | 105 |
| Hamamdagh | Brecciated mud | 0.4 | 0.6 | 0.1 | 6.1 | 14.1 | 352 | 20 | 157 |
| Bendovan | Brecciated mud | 0.4 | 0.8 | 0.1 | 7.4 | 16.2 | 390 | 20 | 142 |
| Khara-Zire | Brecciated mud | 0.3 | 0.6 | 0.1 | 5.7 | 13.4 | 396 | <20 | 101 |
| Garasu | Brecciated mud | 0.5 | 0.6 | 0.1 | 5.8 | 13.9 | 388 | <20 | 113 |

**Table 4.** *Cont.*

| | | | | | | | | | |
|---|---|---|---|---|---|---|---|---|---|
| Hamamdagh | Mud/claystone | 0.2 | 0.6 | 0.1 | 6.0 | 16.6 | 407 | <20 | 138 |
| Hamamdagh | Mud/claystone | 0.2 | 0.6 | 0.1 | 5.9 | 20.1 | 265 | <20 | 132 |
| Bendovan | Mud/claystone | 0.4 | 0.6 | 0.1 | 5.8 | 16.6 | 362 | <20 | 82 |
| Bendovan | Mud/claystone | 0.3 | 0.6 | 0.1 | 6.5 | 18.7 | 307 | 21 | 81 |
| Khara-Zire | Mud/claystone | 0.5 | 0.6 | 0.1 | 5.6 | 17.8 | 311 | <20 | 96 |
| Garasu | Mud/claystone | 0.3 | 0.6 | 0.1 | 5.6 | 16.8 | 452 | <20 | 92 |
| Garasu | Mud/claystone | 0.3 | 0.6 | 0.1 | 6.1 | 15.1 | 403 | 20 | 103 |
| Sangi-Mughan | Mud/claystone | 0.3 | 0.7 | 0.1 | 5.8 | 12.6 | 449 | <20 | 111 |
| Sangi-Mughan | Mud/claystone | 0.5 | 0.7 | 0.0 | 5.3 | 11.1 | 330 | <20 | 98 |
| Bendovan | Sandstone | 0.7 | 0.7 | 0.3 | 5.3 | 14.6 | 352 | <20 | 196 |
| Bendovan | Sandstone | 0.9 | 0.5 | 0.1 | 4.1 | 13.5 | 390 | <20 | 104 |
| Khara-Zire | Sandstone | 0.4 | 0.2 | 0.1 | 2.6 | 12.1 | 292 | <20 | 21 |
| Garasu | Sandstone | 0.2 | 0.5 | 0.1 | 5.0 | 11.0 | 378 | <20 | 132 |
| Sangi-Mughan | Sandstone | 0.0 | 0.6 | 0.1 | 3.7 | 11.3 | 269 | <20 | 363 |
| Zenbil | Liquid mud | 0.3 | 0.6 | 0.1 | 5.3 | 13.3 | n.a. | n.a. | n.a. |
| Bahar | Liquid mud | 0.3 | 0.6 | 0.1 | 5.1 | 14.6 | n.a. | n.a. | n.a. |

| Sampling site | Sample type | Cu ppm | Ni ppm | Rb ppm | Sr ppm | V ppm | Y ppm | Zn ppm | Zr ppm |
|---|---|---|---|---|---|---|---|---|---|
| Hamamdagh | Liquid mud | <20 | 92 | 69 | 415 | 121 | 24 | 29 | 119 |
| Hamamdagh | Liquid mud | 26 | 101 | 72 | 438 | 125 | 28 | 30 | 126 |
| Khara-Zire | Liquid mud | 24 | 90 | 56 | 407 | 100 | 24 | 73 | 128 |
| Garasu | Liquid mud | 24 | 71 | 48 | 368 | 107 | 25 | 33 | 138 |
| Sangi-Mughan | Liquid mud | 21 | 71 | 49 | 362 | 115 | 24 | 29 | 138 |
| Sangi-Mughan | Liquid mud | 20 | 77 | 52 | 654 | 101 | 22 | 30 | 125 |
| Hamamdagh | Oily mud | <20 | 52 | 53 | 330 | 108 | 26 | 26 | 143 |
| Bendovan | Oily mud | 183 | 153 | 42 | 467 | 87 | 37 | 25 | 130 |
| Khara-Zire | Oily mud | <20 | 55 | 58 | 676 | 121 | 24 | 29 | 134 |
| Garasu | Oily mud | 71 | 82 | 73 | 392 | 114 | 30 | 82 | 193 |
| Sangi-Mughan | Oily mud | 26 | 101 | 40 | 521 | 128 | 33 | 22 | 147 |
| Hamamdagh | Brecciated mud | 30 | 95 | 76 | 331 | 122 | 27 | 59 | 134 |
| Bendovan | Brecciated mud | 23 | 100 | 66 | 325 | 124 | 24 | 53 | 137 |
| Khara-Zire | Brecciated mud | 24 | 59 | 79 | 362 | 116 | 26 | 42 | 148 |
| Garasu | Brecciated mud | 53 | 67 | 71 | 344 | 116 | 25 | 84 | 137 |
| Hamamdagh | Mud/claystone | <20 | 95 | 43 | 455 | 123 | 23 | 35 | 132 |
| Hamamdagh | Mud/claystone | 41 | 111 | 65 | 355 | 134 | 28 | 80 | 113 |
| Bendovan | Mud/claystone | <20 | 55 | 76 | 256 | 142 | 26 | 33 | 116 |
| Bendovan | Mud/claystone | <20 | 56 | 70 | 267 | 127 | 26 | 46 | 116 |
| Khara-Zire | Mud/claystone | 37 | 66 | 68 | 238 | 114 | 24 | 39 | 144 |
| Garasu | Mud/claystone | 56 | 100 | 86 | 372 | 110 | 28 | 58 | 135 |
| Garasu | Mud/claystone | 32 | 76 | 59 | 254 | 110 | 24 | 58 | 122 |
| Sangi-Mughan | Mud/claystone | 61 | 70 | 61 | 330 | 107 | 24 | 86 | 132 |
| Sangi-Mughan | Mud/claystone | 45 | 44 | 90 | 140 | 136 | 36 | 93 | 322 |
| Bendovan | Sandstone | <20 | 66 | 38 | 567 | 100 | 25 | 62 | 139 |
| Bendovan | Sandstone | 46 | 62 | 51 | 362 | 79 | <20 | 61 | 115 |
| Khara-Zire | Sandstone | <20 | <20 | 45 | 343 | 20 | 23 | 37 | 123 |
| Garasu | Sandstone | 27 | 62 | 46 | 356 | 86 | 20 | 67 | 104 |
| Sangi-Mughan | Sandstone | 26 | 51 | 42 | 103 | 76 | <20 | 55 | 120 |
| Zenbil | Liquid mud | n.a. | n.a. | n.a. | n.a. | n.a. | n.a. | n.a. | n.a. |
| Bahar | Liquid mud | n.a. | n.a. | n.a. | n.a. | n.a. | n.a. | n.a. | n.a. |

The plot of the chemical data in the A–CN–K diagram illustrates that all sample types fall slightly above the compositional range of Average Proterozoic Shale (APS) and Post-Archean Australian Shale (PAAS), but generally follow the predicted weathering trend for Upper Continental Crust (UCC) rocks (Figure 7). There is also a considerable overlap with the mud compositions reported in Bahar and Zenbil [16], although the mud ejects from all studied localities have slightly lower contents of $Al_2O_3$ and $SiO_2$, which we attribute to a lower clay mineral content (Figure 6). No evidence for sorting (i.e., shift towards the A pole) and K-metasomatism (i.e., shift towards the K pole) is seen. The CIA values are very high compared to basalt (30–45), granite/granodiorite (45–55) and feldspar (50), respectively, i.e., they vary in a range from 66 to 74 for all sample types studied among the different localities (Figure 7). This feature suggests that the solid ejects experience post-depositional

alterations due to fluid–rock–gas reactions and mud diagenesis. Previous studies have shown that these weathering processes can reach a depth of 10 km [5,21,54].

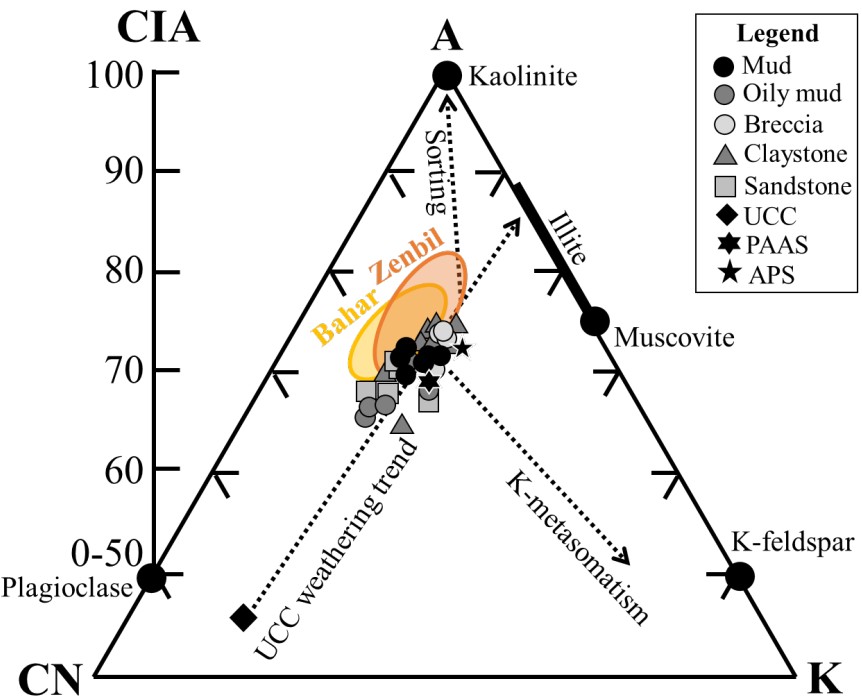

**Figure 7.** Chemical compositions of the mud and sedimentary rocks ejected from the mud volcanoes in Hamamdagh, Bendovan, Garasu, Khara-Zire and Sangi-Mughan, Azerbaijan, plotted in the A–CN–K diagram. The compositions of the Upper Continental Crust (UCC), Post-Archean Australian Shale (PAAS) and Average Proterozoic Shale (APS) are included for comparison. CIA = Chemical Index of Alteration (A < 50 is not shown).

*3.5. Origin of Solid Ejects*

Petrographic, mineralogical and chemical comparisons of all solid ejects from the mud volcanoes at Hamamdagh, Bendovan, Garasu, Khara-Zire and Sangi-Mughan with lithofacies description and the interpretation of several Productive Series outcrops from the South Caspian Basin by Hinds et al. [25] suggests that the majority of the mud, mud/claystones and sandstones are sourced from Pleistocene–Pliocene-aged intervals located in ~1 to 4 km depth, corroborating the calculated fluid generation temperatures (~30 ± 10 °C) and depths (~1–4 km depth). Moreover, Upper Productive Series strata found in several outcrops across the Absheron Peninsula (Azerbaijan), such as sandstones and silt- to mudstones, have mineralogical compositions that resemble those of the lithogenic fragments described in this study [55]. For example, the fined-grained Yasamal Valley sedimentary rocks mainly comprise quartz (15–40 wt.%), feldspar (10–20 wt.%), calcite (10–25 wt.%) and clay minerals (20–60 wt.%), which mineralogically overlap with the mud and mudstones ejected from the five new mud volcanoes (Table 3). Although the sandstone lithologies cropping up in the Yasamal Valley have higher quartz contents (up to 70 wt.%), on average, they are partially pervasively cemented with calcite and minor dolomite (30–40 wt.%), consistent with our petrographic observations made for coarse-grained lithologies [55].

The expelled sandstones and mud/claystones have been interpreted as channelized and sheetflood fluvial deposits and short-lived alluvial plane and lacustrine deposits of a regional Caspian lake, respectively [25,56]. Aliyev et al. [15] argue that these sedimentary rocks are sourced from Jurassic- and Cretaceous-aged, mafic-intermediate, volcanic-to-clastic deposits exposed on the southern slope of the Greater Caucasus (Tufan and Vandam uplifts), as well as from Paleogene- to Miocene-aged, sea and coastal volcanic products and shales of the Gobustan and Absheron peninsulas. The Productive Series sediments of

the South Caspian Basin (especially the sand facies) are further sourced from the Russian platform drained by the paleo-Volga river [56]. Moreover, the integration of the drainage systems (paleo-Volga, paleo-Kura and paleo-Amu Darya) resulted in the extensive supply of sediments from the Russian Platform, Caucasus and Pamir/Kopet-Dagh Mountains, which were deposited as fluvio-lacustrine facies in pre-existing structural depressions of the South Caspian Basin.

The Productive Series strata were deposited during the most dramatic sea level lowstand tract that the Caspian Sea has ever experienced. The extremely high fluvio-deltaic input of the paleo-Volga river also resulted in high depositional rates of the Productive Series sediments of greater than 1.5 km/million years, which ash lasted for at least the last 5 million years [57], producing a ~7–8 km thick sedimentary sequence within the South Caspian Basin that rapidly underwent subsidence and burial diagenesis [58]. Simultaneously, the smaller paleo-Kura and paleo-Amu Darya rivers likely also contributed to the development of the Productive Series strata as well. Rapid burial, post-depositional alterations due to strong fluid–rock–gas reactions and mud diagenesis in a compressive tectonic setting of the South Caspian Basin finally resulted in mud diapirism [16], which expresses in the development of today's landscape of large areas in eastern Azerbaijan and in the formation of new mud volcanic islands within the shallow water regions in the Caspian Sea (Figure 1). Thus, fluid and solid sources for mud volcanic activities are mainly provided by mud chambers to be situated in the Productive Series strata (1 to 5.5 km depth), although a minor contribution from deeper reservoirs (e.g., Maykop Suite shales ranging from 5.5 to 9 km depth and Eocene rocks up to 12 km depth) is likely [21,58,59]. The highly complex architecture of the South Caspian Basin and the mixed origin of the mud volcanic ejects explain the observed chemical and mineralogical compositions of the fluids and mud.

## 4. Conclusions

The geochemical and mineralogical compositions of fluid, mud and rock ejects from five new mud volcanoes located in continental Azerbaijan (Hamamdagh and Bendovan) and in the Caspian Sea (Garasu, Khara-Zire and Sangi-Mughan) were investigated and compared with published compositions of mud and fluids expelled at the mud volcano systems at Bahar (Shamakhi-Gobustan region) and Zenbil (Caspian Sea, Baku Archipelago). The expelled fluids have a Na-Cl composition and are generated via the mixing of evaporated Caspian seawater, shallow 'low-mineralized' pore fluids and deep-seated 'high-mineralized' brines. The fluids contain elevated concentrations of dissolved metal ions (Al, As, Ba, Cu, Li, Si, Sr and Zn: ~60–26,300 µg/L) due to smectite illitization, hydrocarbon maturation and methane production, redox reactions and complex dissolution–precipitation processes involving clay minerals and carbonates. The fluids are generated in Productive Series strata in ~1 to 4 km depth, similar to the mud volcanoes at Bahar and Zenbil (~2 to 4 km depth). The mud is composed of quartz, albite, orthoclase, illite, chlorite, calcite, calcian dolomite, pyrite, Na-smectite and kaolinite. All study sites show lower contents of clay minerals, but higher amounts of quartz and carbonates compared to mud compositions reported in Bahar and Zenbil. However, there is a large mineralogical and chemical overlap of all mud compositions. Chemical element ratios (Ni/Co and Cu/Zn) suggest that the mud forms under anaerobic conditions. The ejected mud, mud/claystones and sandstones are sourced from giant mud chambers located in Productive Series strata, partially fed by deeper reservoirs (Maykop Suite shales and Eocene rocks).

**Author Contributions:** Conceptualization, O.R.A., A.A.A. and E.E.B.; methodology, A.B. (Aygun Bayramova) and A.B. (Andre Baldermann); software, F.M.S. and A.B. (Andre Baldermann); validation, O.R.A., A.A.A., E.E.B. and M.D.; formal analysis, A.B. (Andre Baldermann); investigation, A.B. (Aygun Bayramova); resources, M.D. and A.B. (Andre Baldermann); data curation, A.B. (Aygun Bayramova); writing—original draft preparation, A.B. (Aygun Bayramova) and A.B. (Andre Baldermann); writing—review and editing, A.B. (Aygun Bayramova) and A.B. (Andre Baldermann); visualization, A.B. (Andre Baldermann); supervision, O.R.A., A.A.A. and E.E.B.; project adminis-

tration, A.B. (Andre Baldermann); funding acquisition, A.B. (Aygun Bayramova) and A.B. (Andre Baldermann). All authors have read and agreed to the published version of the manuscript.

**Funding:** This research was partly funded by the NAWI Graz Geocenter (Graz University of Technology) and supported further by the TU Graz Open Access Publishing Fund.

**Data Availability Statement:** All raw data will be made available upon request to A.B.

**Acknowledgments:** The authors are grateful to S. Eichinger, J. Jernej, M. Hierz, A. Wolf and S. Perchthold (IAG, TU Graz) for their analytical support.

**Conflicts of Interest:** The authors declare no conflict of interest.

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
