# Peer review of "Tracing Water–Rock–Gas Reactions in Shallow Productive Mud Chambers of Active Mud Volcanoes in the Caspian Sea Region (Azerbaijan)"

_minerals, doi:10.3390/min13050696_

Round 1
Reviewer 1 Report
Reviewer comments
Manuscript ID: minerals-2404048
Authors: Aygun Bayramova, Orhan R. Abbasov, Adil A. Aliyev, Elnur E. Baloglanov, Franziska M. Stamm, Martin Dietzel and Andre Baldermann
Title: Tracing water-rock-gas reactions in shallow productive mud chambers of active mud volcanoes in the Caspian Sea region (Azerbaijan)
Bayramova et al. study fluids and mud erupted by onshore (continental Azerbaijan) and offshore (Caspian Sea) mud volcanoes, reconstruct fluid generation depths and mud provenance. The reported results are of interest for the scientific community, especially, in the field of generation and migration of deep fluids and mud volcanism.
General comments:
The paper is well structured and presents interesting novel results relevant to the objectives of the study. The manuscript can be accepted for publication after a moderate revision. Please find specific comments below.
1. It is unclear how clay minerals are identified. XRD analysis alone cannot provide precise estimates of the relative percentages of illite and smectite. Was the clay mineral fraction (< 2 μm) separated? Were heated and glycolated specimens analyzed? It is also difficult to determine illite crystallinity from analysis of bulk samples. Please, describe the methods used to identify clay minerals.
2. More caution is required in classifying minerals in the mud as authigenic or detrital because the manuscript lacks details of mineral morphology (SEM data) and analysis of stable isotopes. Namely, in section 3.3 (lines 273-275), illite is unsubstantially interpreted as a detrital phase, but it could form by illitization of smectite during late burial diagenesis of mudrocks. Furthermore, the phrase in Conclusions saying that ‘The fluids contain elevated concentration of dissolved metal ions (Al, As, Ba, Cu, Li, Si, Sr and Zn: ~60-26300 μg/l) due to smectite-illitization…’ (lines 388-389) contradicts the statement a few lines below on ‘minor contributions from deeper reservoirs (Maykop Suite shales and Eocene rocks)’ in mud volcanic products (lines 398-400). Thus, it remains unclear whether illite authigenic or detrital.
The authigenic origin of dolomite is supported by only two facts: its low ordering degree and Ca-excess (section 3.3, lines 270-271). The authors also refer to fluid chemical data, but a positive saturation index (SIdolomite) itself is insufficient to ensure the crystallization of dolomite from such a solution as the only option (see Wang, 2021 DOI 10.1016/j.marchem.2021.104017). To be interpreted unambiguously as authigenic or detrital, dolomite should be characterized in terms of grain morphology, relationships with other minerals, and carbon isotope composition (δ13С), but no such data are provided in the manuscript. Please, either give more convincing data or formulate your conclusions as hypothetical.
3. Providing more information on the phase composition of sediments in the Productive Series strata, which the authors consider to be the main source of fluids and solid matter (section 3.5 ‘Origin of solid ejects’), will allow explicit comparison of the sediments with the erupted mud.
4. The fluid generation temperature is estimated in the manuscript using three geothermometers (Na-K, K-Mg, and SiO2) while the values obtained with the K-Mg geothermometer are mentioned as overestimated due to ‘removal of Mg2+ ions from the fluids through the precipitation of dolomite and smectite minerals’ (lines 251-252). However, 1-2 % of dolomite in a half of mud samples hardly could affect considerably the pore fluid composition (also, see the comments on the authigenic origin of dolomite above, point 2). On the other hand, the fluids are described as mixed, with contributions from the evaporated Caspian seawater and low- to high-salinity pore waters (Fig. 5), but the presence of seawater containing times more Mg than K would make the Mg/K ratios in the fluid much higher. Are there any other explanations why the K-Mg geothermometry results may be overestimated?
The inferred minor contributions from deeper reservoirs (Maykop Suite shales and Eocene rocks) (Conclusion, lines 398-400) may indicate that K-Mg geothermometry records a contribution of a higher-temperature fluid component. Note that the Mg-Li geothermometer has been used successfully for reconstructing fluid generation temperatures in mud volcanic systems (Lavrushin et al., 2003 DOI 10.1023/A:1023452025440, 2021 DOI 10.1134/s0024490221060043; Kikvadze at al., 2019 DOI 10.1007/s11707-019-0810-8). Please, add Mg-Li geothermometry and discuss the results.
Some specific comments and suggestions:
Table 1. Please, add TDS (total dissolved solids) values.
Table 3. The table shows data for muscovite in the mud and sedimentary rocks. Did you mean illite instead? Please, correct.
Lines 308-310: ‘As for all eject types, the positive correlation of rubidium (Rb) and vanadium (V) with Al2O3, yttrium (Y) and zircon (Zr), as well as TiO2 and Ni and SO3 suggests their association with clay minerals, heavy minerals and sulfide minerals, respectively [41].’ Please, add the correlation coefficient values.

Author Response
Dear respected reviewer,
please find our detailed response to your criticism in the attached file.
Kind regards,
Andre Baldermann

Reviewer 2 Report
This paper reports mineralogical and chemical data on several new mud volcanoes located in the east region of the Caspian Sea coast. The authors analysed both liquid and solid components of the mud volcanoes and discuss the possible origin in light of shallow mud chambers in the oil productive series. The work is on interest for the Minerals’s reader and I recommend publication. However, several points appear confusing to me and should be take into account in the new version of the manuscript.
Fluid saturation state was estimated with PHREEQC (line 143). To do this calculation, the fluid silica concentration is need. This is lucking in Table 1 and should be added, otherwise estimations on alumina silicate saturation state are meaningless. In addition, fluid alkalinity should be reported as you affirmed to be measured it (line 11). Another main point on the saturation state of fluids are the Al data. Filtering through 0.45 possible Al-colloidal fractions (often Al-oxy-hydroxides) may pass through. This produces a possible overestimation of dissolved Al concentrations saturation state with respect to the aluminum-silicate minerals. An additional problem is the conditions of applicability of your thermometric equations (1,2,3). To be applicable, the mineral-fluid equilibria must be evaluated. The classical way is through the fluid saturation index. Mineralogical estimation of detrital mad constituents are mainly clastic products as reported in the mad origin section (Fig. 7).Author Response
Dear respected reviewer,
please find our detailed response to your criticism in the attached file.
Kind regards,
Andre Baldermann
